# Chitosan-Strontium Oxide Nanocomposite: Preparation, Characterization, and Catalytic Potency in Thiadiazoles Synthesis

**DOI:** 10.3390/polym14142827

**Published:** 2022-07-12

**Authors:** Khaled D. Khalil, Sayed M. Riyadh, Nazeeha S. Alkayal, Ali H. Bashal, Khadijah H. Alharbi, Walaa Alharbi

**Affiliations:** 1Department of Chemistry, Faculty of Science, Cairo University, Giza 12613, Egypt; riyadh1993@hotmail.com; 2Department of Chemistry, Faculty of Science, Taibah University, Al-Madinah Almunawarah, Yanbu 46423, Saudi Arabia; abishil@taibahu.edu.sa; 3Department of Chemistry, Faculty of Science, Taibah University, Al-Madinah Almunawarah 30002, Saudi Arabia; 4Chemistry Department, Faculty of Science, King Abdulaziz University, P.O. Box 80203, Jeddah 21589, Saudi Arabia; nalkayal@kau.edu.sa; 5Department of Chemistry, Science and Arts College, Rabigh Campus, King Abdulaziz University, P.O. Box 80203, Jeddah 21589, Saudi Arabia; khalharbe@kau.edu.sa (K.H.A.); wnhalharbe@kau.edu.sa (W.A.)

**Keywords:** strontium oxide, chitosan, nanocomposite film, 2-hydrazono [1,3,4]thiadiazole, heterogeneous catalysis

## Abstract

Recently, Strontium oxide (SrO) nanoparticles (NPs) and hybrids outperformed older commercial catalysts in terms of catalytic performance. Herein, we present a microwave-assisted easy in situ solution casting approach for the manufacture of strontium oxide nanoparticles doped within a naturally occurring polymer, chitosan (CS), at varying weight percentages (2.5, 5, 10, 15, and 20 wt.% SrO/chitosan). To construct the new hybrid material as a thin film, the produced nanocomposite solutions were cast in petri dishes. The aim of the research was to synthesize these hybrid nanocomposites, characterize them, and evaluate their catalytic potential in a variety of organic processes. The strontium oxide-chitosan nanocomposites were characterized using Fourier transform infrared (FTIR), X-ray diffraction (XRD), and scanning electron microscope (SEM) techniques. All the results confirmed the formation of chitosan–strontium oxide nanocomposite. FTIR spectrum of nanocomposite showed the presence of a characteristic peak of Sr-O bond. Furthermore, XRD revealed that SrO treatment increased the crystallinity of chitosan. The particle size was calculated using the Debye–Scherrer formula, and it was determined to be around 36 nm. The CS-SrO nanocomposite has been proven to be a highly efficient base promoter for the synthesis of 2-hydrazono [1,3,4]thiadiazole derivatives. To optimize the catalytic method, the reaction factors were investigated. The approach has various advantages, including higher reaction yields, shorter reaction durations, and milder reaction conditions, as well as the catalyst’s reusability for several applications.

## 1. Introduction

With the increasing interest in nanotechnology, many researchers are concentrating their efforts on the development of metal oxide nanoparticles for a variety of applications [1,2,3]. The acquisition of novel chemical and physical properties distinguishes advancements in nanomaterials’ synthesis pathways.

Natural polysaccharides have been widely used as a templates for stabilizing and immobilizing metal oxide nanoparticles in recent years [4,5,6]. Chitosan (CS), a partially deacetylated version of chitin, is regarded a good stabilizer to be efficiently capped with these metal oxide nanoparticles due to its unique structural features, including the presence of numerous hydroxyl and amino groups [7,8,9]. Strontium oxide is recognized as an excellent catalyst for transesterifications and is one of the most promising heterogeneous base catalysts among processable alkaline earth metal oxides, with the highest catalytic activity in studies [10,11]. Although calcium and magnesium oxides are more widely used than strontium oxide, the latter performs better. These limitations are due to the arduous preparation and size control problems, and extensive research is required to make SrO-based catalysts economically favorable. To overcome this limitation, many researchers are now attempting to synthesize strontium oxide (SrO) nanoparticles using simple methods, such as wet and bio-reduction methods [12,13], and investigating their properties and utility in a variety of applications, including gas sensor electrodes, lithium-ion batteries, semiconductors, and super capacitors [12,13,14,15,16].

For the aforementioned reasons, we attempted to use the chitosan-strontium oxide nanocomposite in this study to synthesize 1,3,4-thiadiazoles. As the latter mentioned compounds, 1,3,4-thiadiazole derivatives, have been characterized with unique properties that fit in many applications such as antibacterial, antifungal, antihepatitic, antioxidant, antileishmanial, anti-inflammatory, analgesic, anticancer, antidiabetic, antihypertensive, analgesic, diuretic, central nervous system (CNS) depressing, and anticonvulsant effects, according to a literature review [17,18,19,20,21]. Since the synthesis of these thiadiazoles involves the presence of base catalyst, it is worthwhile to determine if they could be conducted in the presence of CS-SrO nanocomposite.

Thus, as a continuation of our previous research in the field of green synthesis reactions, we present this eco-friendly technique to overcome the technical problems associated with the old harmful catalysts, we have prepared and characterized chitosan capped with strontium oxide nanoparticles (Figure 1) as a promising hybrid nanocomposite, and then investigate its catalytic efficiency in the production of 2-hydrazono [1,3,4]thiadiazole derivatives.

## 2. Materials and Methods

Chitosan was provided by Sigma Aldrich (powder, medium molecular weight, shrimp shells source, batch no. C3646, density = 0.15–0.3 g/cm^3^) and SrO (powder, product no. 415138). Potassium hydroxide, water, methanol, and acetic acid, were purchased from Merck Company and was used as such without further purification. Fourier transform infrared spectra (FTIR) with a Nicolet Magna 6700 FT spectrometer (Thermo Fisher Scientific, Waltham, MA, USA) were conducted in a wavenumber region (500–4000 cm^−1^). A Varian Mercury VXR-300 spectrometer (Varian, Willich, Germany) was used to record the ^1^H and ^13^C NMR spectra (300 MHz for ^1^H NMR and 75 MHz for ^13^C NMR) and the chemical shifts were correlated to the DMSO-*d*_6_ solvent. The mass spectra were measured on a GCMSQ1000-EX Shimadzu and GCMS 5988-A HP spectrometers (Shimadzu, Kyoto, Japan) with a 70-eV ionizing voltage. Melting points of the synthesized compounds were obtained using an electrothermal Gallenkamp equipment (GallenKamp, Lister, UK) and are uncorrected. X-ray diffraction (XRD) patterns were studied using a Philips diffractometer (Model: X’Pert-Pro MPD; Philips, now PANaytical, Malvern, Worcestershire, UK) with Cu Kα radiation (wavelength 1.5418 Å) at 40 kV and 40 mA. The patterns were collected between 2θ of 5° and 60°, and the scan speed was 1.5 degree/min. For SEM and EDX (HRSEM, JSM 6510A, Jeol Ltd., Tokyo, Japan) measurements, the thin films were cut into small pieces and put on the SEM stubs with carbon tape. Then the samples were coated with 4 nm thickness of platinum layer, after that transferred into SEM Teneo/Quattro for imaging. Images were taken under high vacuum with different magnifications. A 2-{1-[4-(2,4-dihydroxyphenylazo)phenyl]ethylidene} thiosemicarbazide (1) [22] and N-aryl arenecarbohydrazonoyl halides **2a**–**e** [23,24] were prepared as reported in literature.

### 2.1. CS-SrO Nanocomposite Film Preparation

Modified microwave aided solution casting was used to synthesize chitosan-Strontium (CS/SrO) nanocomposite films [25,26,27]. Dissolving medium molecular weight chitosan in a 2 percent (*w*/*v*) aqueous acetic acid solution for 48 h at room temperature yielded a 2 wt.% chitosan solution. To create a homogeneous transparent chitosan solution, the viscous solution was filtered using 90 mm Whatman filter paper. A portion of this solution was placed in a 50 mL bottle, and portions of 5, 10, 15, and 20 (*w*/*v* percent) SrO were added portion by portion under vigorous stirring, and the stirring was continued for another 24 h. In order to obtain the best results, the mixture was then microwaved for different time intervals 3, 4, and 5 min and under different powers at 300, 400, and 500 watts. The solution was put into a Teflon petri dish (8 cm) and dried in a vacuum oven set to 50 °C for 3 days to evaporate the solvent. The chitosan-SrO nanocomposite film was pulled off the petri dish and soaked in distilled water after being neutralized with 5 mL of 1 M KOH. Finally, the film was kept in a vacuum desiccator for two days at ambient temperature.

### 2.2. Reactions of 2-{1-[4-(2,4-Dihydroxyphenylazo)phenyl]ethylidene}thiosemicarbazide (1) with N-Aryl Arenecarbohydrazonyl Halides ***2a**–**e***


**Method A**


In 10 mL dioxane containing catalytic amount of triethylamine (0.1 g), a mixture of 2-[4-(2,4-dihydroxyphenylazo)phenyl]ethylidenethiosemicarbazide (1) (0.329 g, 1 mmol) and appropriate N-aryl arenecarbohydrazonoyl halides **2a**–**e** (1 mmol) was refluxed for 3–4 h till all the starting material was consumed (as monitored by TLC). Under reduced pressure, excess solvent was removed, and the reaction mixture was triturated with methanol. The separated product was then filtered, washed with methanol, dried, and finally recrystallized from ethanol to give products **5a**–**e**.


**Method B**


Unlike technique A, the reaction mixture in this process was subjected to MW radiation at 300 Watt in a closed Teflon tank until all of the starting material had been consumed.


**Method C**


In 10 mL dioxane containing 0.1 g of CS-SrO film, a mixture of 2-[4-(2,4-dihydroxyphenylazo)phenyl]-ethylidenethiosemicarbazide (1) (0.329 g, 1 mmol) and suitable N-aryl arenecarbohydrazonoyl halides **2a**–**e** (1 mmol) was refluxed for 3–4 h until all the starting material was consumed (as monitored by TLC). The CS-SrO film was carefully removed by filtration and rinsed with hot ethanol once the reaction was completed. Under reduced pressure, excess solvent was removed, and the reaction mixture was triturated with methanol as usual. The separated product was then filtered, washed with methanol, dried, and finally recrystallized from ethanol to give products **5a**–**e**.


**Method D**


Unlike technique C, the reaction mixture in this method was subjected to MW radiation at 300 Watt in a closed Teflon tank until all of the starting material had been consumed.


**2-{2-[1-(4-(2,4-Dihydroxyphenylazo)phenyl)ethylidene]hydrazono}-3,5-diphenyl-2,3-dihydro-[1,3,4]thiadiazole (5a)**


Dark red powder; mp 198–200 °C; IR (KBr) *n*_max_ 3418 (br, 2OH), 1598 (C=N) cm^−1^; ^1^H-NMR (DMSO-*d*_6,_ 300 MHz): *d* = 2.31 (3H, s, CH_3_-C=N-N), 7.04–8.17 (17H, m, Ar-H ), 6.50 (1H, s, OH), 6.55 (1H, s, OH); ^13^C-NMR (DMSO-*d*_6,_ 75 MHz): *d* = 16.1 (CH_3_-C=N-N), 103.1, 108.7, 121.4, 122.1, 123.4, 124.8, 125.8, 127.4, 129.7, 131.7, 132.7, 133.8, 138.4, 140.4, 143.3, 149.5, 153.5, 156.6, 157.4, 160.3, 163.6; EIMS *m*/*z* (%): 506 [M^+^] (60), 77 (100). Anal. Calcd for C_28_H_22_N_6_O_2_S (506.15): C, 66.39; H, 4.38; N, 16.56; S, 6.33. Found: C, 66.56; H, 4.52; N, 16.44; S, 6.49.


**2-{2-[1-(4-(2,4-Dihydroxyphenylazo)phenyl)ethylidene]hydrazono}-3-phenyl-5-(4-methylphenyl)-2,3-dihydro-[1,3,4]thiadiazole (**
**5b)**


Dark red powder; mp 210–212 °C [22]


**2-{2-[1-(4-(2,4-Dihydroxyphenylazo)phenyl)ethylidene]hydrazono}-3-phenyl-5-(4-methoxy phenyl)-2,3-dihydro-[1,3,4]thiadiazole (5c)**


Dark red powder; mp 200–202 °C [22]


**2-{2-[1-(4-(2,4-Dihydroxyphenylazo)phenyl)ethylidene]hydrazono}-3-phenyl-5-(4-chlorophenyl)-2,3-dihydro-[1,3,4]thiadiazole (5d)**


Dark red powder; mp 216–218 °C; IR (KBr) *n*_max_ 3412 (br, 2OH), 1596 (C=N) cm^-1^; ^1^H-NMR (DMSO-*d*_6,_ 300 MHz): *d* = 2.34 (3H, s, CH_3_-C=N-N), 7.08–8.19 (16H, m, Ar-H ), 6.53 (1H, s, OH), 6.58 (1H, s, OH); ^13^C-NMR (DMSO-*d*_6,_ 75 MHz): *d* = 15.9 (CH_3_-C=N-N), 103.2, 117.7, 121.4, 122.7, 123.6, 124.8, 125.8, 127.4, 129.6, 131.7, 132.7, 134.8, 138.4, 141.4, 145.3, 149.5, 153.5, 156.6, 157.4, 160.3, 163.5; EIMS *m*/*z* (%): 540 [M^+^] (40), 77 (100). Anal. Calcd for C_28_H_21_ClN_6_O_2_S (540.11): C, 62.16; H, 3.91; N, 15.53; S, 5.93. Found: C, 62.06; H, 4.02; N, 15.44; S, 6.09.


**2-{2-[1-(4-(2,4-Dihydroxyphenylazo)phenyl)ethylidene]hydrazono}-3-(4-nitrophenyl)-5-phenyl-2,3-dihydro-[1,3,4]thiadiazole (5e)**


Dark red powder; mp 192–194 °C [22]

## 3. Results and Discussion

### 3.1. Characterization of Chitosan-Strontium Oxide Nanocomposite Film

#### 3.1.1. Characterization via FTIR

Figure 2 shows a comparable FTIR examination of native chitosan (A) and chitosan-strontium oxide (B) nanocomposite. The chitosan spectrum (A) showed the broad stretching band at u = 3408 cm^−1^, due to the OH, and NH_2_ overlapped stretching bands that lie in the same region [7,8]. Moreover, the usual characteristic bands of chitosan manifested at u = 1658 and 1609 cm^−1^ (for CONH, amide group band), and those of aliphatic CH appeared at 2918, 2875 cm^−1^. On the other hand, Figure (B) showed the FTIR of the hybrid chitosan-SrO nanocomposite, there are obvious changes especially in the fingerprint region. As reported in the literature, the SrO NPs has peaks between 500 and 1000 cm^−1^ at 733.23 cm^−1^, 810.10 cm^−1^, and 856.39 cm^−1^ that can be directly traced to Sr-O bending vibrations [28], additional peak at 854.35 cm^−1^ is considered as clear evidence for the incorporation of strontium oxide and its coordination with the binding sites along chitosan backbone.

#### 3.1.2. SEM and Morphological Characteristics

In order to explore the morphological changes that occurred to chitosan surface of chitosan upon the interaction with SrO molecules, Figure 3 shows the FESEM micrographs for chitosan, strontium oxide nanoparticles, and chitosan strontium oxide composite. The micrograph of the native chitosan (A) explored the usual non-porous, fibrous surface as previously reported in the literature [7,8,9]. In Figure 3B, the image of SrO nanoparticles showed spherical shape and some aggregations as reported in the literature [29].

Subsequently, the CS-SrO nanocomposite (C) showed a clear congregations appeared distributed in certain regions which are ascribed to the action with the SrO molecules in these areas which means that the accumulation of SrO could be achieved and enhanced.

#### 3.1.3. Energy-Dispersive X-ray Spectroscopy (EDS) and Estimation of Strontium Amount

Figure 4 shows an EDS graph of chitosan-SrO nanocomposites that was used to quantify the strontium content within the chitosan. The hybrid material’s EDS showed the appearance of the standard Sr signals, which confirmed its incorporation inside the polymer. Sr content was 13.06 wt.%, as seen in Figure 4.

#### 3.1.4. X-ray Diffraction Pattern (XRD)

Structural lineaments of the unmodified chitosan (A), and chitosan-strontium oxide composites with 10 wt.% (B), were explored by measuring X-ray diffraction technique as seen below in Figure 5. In Figure 5A, the usual peak of chitosan appeared at 2θ (16–22°), which in agreement with the literature values of the hydrated crystalline structure of chitosan [25,26]. In Figure 5B, the XRD pattern showed the same characteristic peak of chitosan but with slight depression as result of the coordination between strontium oxide molecules and chitosan chain. In addition, the diffractogram of synthesized chitosan-strontium oxide catalyst confirms that SrO molecules were successfully deposited on the chitosan chain. The crystallinity of SrO was maintained in the nanocomposite, with minor shift in the detected peaks as compared to that reported for commercial SrO. The presence of strontium oxide molecules is responsible for the additional peaks shown in the diffractogram at 2θ = 26.46, 30.02, 32.36, 38.25, 44.14, 48.26, and 50.21 corresponding to the SrO cubic structure’s reflection planes [30,31]. However, crystallinity was improved to some extent in other regions due to the incorporation of strontium oxide molecules with the active sites (NH_2_ and OH) along the chitosan backbone. Using the following Debye–Scherrer formula (1) [32], the average grain size was estimated to be 36 nm.
(1)$$D\left(\mathrm{nm}\right)=\frac{0.9*\lambda }{\beta \cdot \cos\theta }$$
where, D(nm): is the crystalline size in nm, β can be calculated for the most intense peak for CS-SrO nanocomposite pattern at λ which is the wavelength of Cu-kα1 = 1.54060 A°.

### 3.2. Synthesis of 2-Hydrazono [1,3,4]thiadiazole Derivatives

In order to investigate the nanocatalyst’s capability to catalyze the cyclocondensation reaction of 2-{1-[4-(2,4-dihydroxyphenylazo)phenyl]ethylidene}thiosemicarbazide **1** with *N*-aryl arene-carbohydrazonoyl halides **2a**, the reaction was conducted in the presence of base catalysts (TEA or CS-SrO nanocatalyst) under thermal and microwave conditions (Figure 1).

#### 3.2.1. Catalyst Loading and Reaction Conditions Optimization

A model reaction of thiosemicarbazide 1 with carbohydra-zonoyl halides **2a** was carried out using 2.5, 5, 10, 15, and 20 wt.% of nanocomposite film under the same circumstances to estimate the suitable catalyst loading. Based on the findings, a catalyst loading of 10 wt.% was shown to be the most efficient quantity for achieving maximal reaction progress (95% yield) after 20 min of microwave irradiation (300 Watt) (Figure 6). Moreover, The recovered catalyst was effectively utilized four times without substantial catalytic potency change (Figure 7). The Appendix A are provided at the end of the article and contain all the analytical data for chemical **5a**. 

#### 3.2.2. Utility of CS-SrO as Nanocatalyst in the Synthesis of -1,3,4-Thiadiazole Derivatives

Now, an environ-economic synthesis of 2-hydrazono-1,3,4-thiadiazole derivatives was successfully achieved. Thus, cyclocondensation of 2-{1-[4-(2,4-dihydroxyphenylazo)phenyl]ethylidene}thiosemicarbazide (**1**) with *N*-aryl arene-carbohydrazonoyl halides **2a**–**e** resulted in the formation of target thiadiazoles **5a**–**e**. The effectiveness of the process was verified by comparing the yield percentage of the products obtained with triethylamine as harsh basic reagent and chitosan-SrO nanocomposite as a powerful, ecofriendly, base catalyst under both thermal and microwave conditions (Figure 2) (Table 1).

The effect of catalyst and heating conditions was examined with various substituents in aromatic ring of hydrazonoyl halides **2a**–**e** (Table 1). The results indicated that, under thermal condition, triethylamine provided good yield (68–72%) with traces of starting materials observed on TLC. The yields were improved with CS-SrO nanocatalyst (81–84%) (Table 1). Encouraged by these results, we have investigated this cyclo-condensation reactions under microwave irradiation. As shown in Table 1, nanofiber act as superior basic catalyst to furnish the isolated products **5a**–**e** in excellent yields (91–95%) without formation of any side-products in comparison to triethylamine (73–78%) under the employed conditions.

## 4. Conclusions

In this study, chitosan-Strontium oxide nanocomposite films were prepared, with different weight percentages ranging from 2.5 to 20 wt.%, using the microwave-assisted easy in situ solution casting approach. The nanocomposite film was carefully studied using FTIR, XRD, FESEM, and EDS measurements. All of the tools’ findings indicated the presence of strontium oxide molecules within the chitosan matrix. FTIR spectra showed an obvious change especially in the finger print region~500 and 1000 cm^−1^ which can be traced back to bending vibrations of strontium oxide. In addition, in the XRD pattern, a combination of chitosan and SrO characteristic peaks could be seen. Moreover, the nanocomposite revealed a clear uniform surface alteration of chitosan after coordination with SrO molecules. In the synthesis of 1,3,4-thiadiazole derivatives, this CS-SrO nanocomposite was successfully employed as an eco-friendly heterogeneous basic catalyst. Because of its environmental and economic impact, the nanocatalyst might be employed in the industrial manufacturing of the heterocyclic compounds. Finally, we conclude that the nanocomposite base catalyst can be used to efficiently synthesize a wide range of heterocycles that were previously generated via non-green methodologies.

## Data Availability

Not applicable.

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
