# Peer review of "Chitosan-Strontium Oxide Nanocomposite: Preparation, Characterization, and Catalytic Potency in Thiadiazoles Synthesis"

_polymers, 2022, doi:10.3390/polym14142827_

Round 1
Reviewer 1 Report
In this manuscript, the authors prepared chitosan-strontium oxide nanocomposites and tested their catalytic performances. I would suggest the acceptance of the manuscript after the following revision:
1. The introduction section needs to be reorganized.
(1) Some transition sentences are needed before and after Lines 59-62 to connect with the previous and following paragraphs.
(2) Lines 54-55, "To overcome this limitation, many researchers are now attempting to synthesize strontium oxide (SrO) nanoparticles using simple methods." What are the simple methods specifically? Since the aim of the research is to synthesize and characterize the hybrid nanocomposites. I suggest the authors to specify what methods have already been applied to synthesize these nanocomposites.
(3) Some references about the synthesis of other polymer/nanoparticle nanoscomposites can also be cited: "Core/shell conjugated polymer/quantum dot composite nanofibers through orthogonal non-covalent interactions." Polymers 8.12 (2016): 408.; "Polymer-nanoparticle composites: from synthesis to modern applications." Materials 3.6 (2010): 3468-3517.; "Linear and nonlinear behavior of crosslinked chitosan/N-doped graphene quantum dot nanocomposite films in cadmium cation uptake." Science of The Total Environment 690 (2019): 1245-1253.
2. The materials and methods section also need some revision. The transition to lines 133-159 can be a little bit abrupt, where are the NMR spectra? The description of methods A-D are also not clear. Method A says "the separated product was then filtered, washed with methanol, dried, and finally recrystallized from ethanol to give products 5a-e." And the same sentence also appears in Method C. So which method was used to synthesize 5a-e?
3. I suggest the authors to label the peaks mentioned in the manuscript in Figure 2 to help the readers to identify what they should look at.
4. I suggest the authors to also add higher magnification SEM images to Figure 3.
5. Calculation based on Scherrer equation shows that the average grain size is 36 nm. I suggest the authors to image the particles using TEM or higher magnification SEM to characterize the size and uniformity of the particles.
6. Table1, i suggest the authors to also compare the yields of the catalysts after several recycles. Also I suggest the authors to perform XRD analysis of the recycled catalyst to see if the crystal structures are maintained.
Author Response
Responses to the reviewer comments
Authors would like to thank the reviewer for his valuable comments.
Please see the following table of our responses, corrections and explanation based on the reviewer comments.
|
Reviewer 1 Comments |
Our Reply |
1 |
Some transition sentences are needed before and after Lines 59-62 to connect with the previous and following paragraphs. |
Thank you for your comment, additional transition sentences are added in the introduction part. |
2 |
Lines 54-55, "To overcome this limitation, many researchers are now attempting to synthesize strontium oxide (SrO) nanoparticles using simple methods." What are the simple methods specifically? Since the aim of the research is to synthesize and characterize the hybrid nanocomposites. I suggest the authors to specify what methods have already been applied to synthesize these nanocomposites. |
Although the methods mentioned in the references [11-14], but additional reference [11] is added to clarify this point. |
3 |
Some references about the synthesis of other polymer/nanoparticle nanoscomposites can also be cited: "Core/shell conjugated polymer/quantum dot composite nanofibers through orthogonal non-covalent interactions." Polymers 8.12 (2016): 408.; "Polymer-nanoparticle composites: from synthesis to modern applications." Materials 3.6 (2010): 3468-3517.; "Linear and nonlinear behavior of crosslinked chitosan/N-doped graphene quantum dot nanocomposite films in cadmium cation uptake." Science of The Total Environment 690 (2019): 1245-1253. |
The first and third references are not related to our interest in this article, as herein we consider only the metal oxide nanoparticles within the polymer matrix and their catalytic potency. Only the second one is a good additive and cited in the introduction part. |
4 |
The materials and methods section also need some revision. The transition to lines 133-159 can be a little bit abrupt, where are the NMR spectra? The description of methods A-D are also not clear. Method A says "the separated product was then filtered, washed with methanol, dried, and finally recrystallized from ethanol to give products 5a-e." And the same sentence also appears in Method C. So which method was used to synthesize 5a-e? |
In method A and B, triethyl amine (TEA) is used as catalyst, but in method A under thermal conditions and in method B under microwave irradiation. In the C and D, the above methods were repeated but in presence of CS-SrO nanocatalyst instead of the traditional catalyst (TEA). Anyhow, the sentences revised and modified with more explanation. |
5 |
I suggest the authors to label the peaks mentioned in the manuscript in Figure 2 to help the readers to identify what they should look at. |
Unfortunately, labelling the peaks in the figure will change the resolution of the image. |
6 |
I suggest the authors to also add higher magnification SEM images to Figure 3. |
A original SEM images were sent to the editorial office just after the article submission. |
7 |
Calculation based on Scherrer equation shows that the average grain size is 36 nm. I suggest the authors to image the particles using TEM or higher magnification SEM to characterize the size and uniformity of the particles. |
Unfortunately, TEM is not available to measure because the instrument under maintenance. In fact, the Scherrer equation gave only an approximate value. |
8 |
Table1, I suggest the authors to also compare the yields of the catalysts after several recycles. Also, I suggest the authors to perform XRD analysis of the recycled catalyst to see if the crystal structures are maintained. |
Thank you for your comment, actually we compared the results of the recycled catalyst for a model reaction 5a, as the reaction in all alternatives is the same. Moreover, the XRD for the recycled catalyst didn’t show any dramatic changes in the structure which is in agreement to the results of the studied catalytic reactions. |

Reviewer 2 Report
A nanocomposite thin films of Chitosan/Strontium oxide (SrO) (different weight percentage) have been prepared using microwave assisted solution casting technique. The structural behavior of the samples along with the catalytic ability were explored using various techniques. This research is up to date and it is consistence with the general scope of the journal. However, some modifications should be considered by the authors before accepting the manuscript for publication in Polymers. Detailed comments are given below:
1- All abbreviations should be first defined then used throughout the manuscript. For example, SrO in “Abstract” section, or CNS depressing (page 2, line 61) and some other cases in the manuscript.
2- In the introduction section the authors need to put more effort to clearly address the gap in the current researches in this area and why carrying out this research is important. In other words, what is the research questions the authors have? I feel the authors should clearly state what are the reasons for performing the study. They can add this to the last paragraph of the introduction.
Furthermore, the third paragraph (page 2, line 59) is not well-structured, and there is no topic sentence that should give the reader a clue about the entire paragraph. Thus, it should be modified and expanded with more recent works from the literature.
3- Figure 1 is placed in the introduction section without been addressed or explained in the text. This should be fixed.
4- In Materials and Methods section the molecular weight of used raw material like (CS) should be added, as it significantly impacts on the overall behavior of used host medium.
5- In Materials and Methods section the range of FTIR is specified to be (400 – 4,000 cm−1) while in the section (3.2.1. Characterization via FTIR) (page 5) the range is different and it should be corrected. Similar mistake should be corrected for the XRD were the range is mentioned to be 2θ of 10° and 80°; while in Figure 5 the range is from 0° and 60° and this should be corrected.
6- In page 5 line 88, what is the meaning of 4nm Pt? This should be clarified.
7- All used equations should be numbered in sequence and cited (such equation in page 6 line 220).
8- In page 3 line 96 authors stated that “the viscous solution was filtered” This statement needs to be more clarified by addressing the filtering process and what has been used as a filter.
9- There are some grammatical, spelling, format and punctuation mistakes, please revise the manuscript accurately (for instant page 3 line 100 the word “diffeerent”, and some other cases in the manuscript.
10- What is the purpose of soaking the films back in distill water after film formation, as stated in page 3 line 103? How author confirm that there is no dissolution of the film during this process? This should be clarified.
11- Some citation colors are different and not accurately include the information mentioned in the sentences; for instant, page 4 line (163, 173), page 5 line (188), page 6 line (219).
12- Section (3.1. Preparation of chitosan-strontium oxide nanocomposite film) should be moved to the Material and Method section, then all the section should be numbered in sequence.
13- In page 4 line 168 “Figure 1 shows” should be changed to Figure 2 as the FTIR is presented in Figure 2.
14- The citation style should be unified as the one written in page 4 line 171 [(Cardenas and Miranda, 2004)].
15- In page 4 line 172 it is written u = 1658 cm-1? What is u? It should be defined.
16- In Figure 2 the FTIR spectra for all other concentration of SrO should be added in order to clarify the impact of nanoparticle concentration on the structural behavior of the prepared thin films. Similarly, the SEM and XRD for all other concentration are necessary to be added.
17- In page 5 line 188, what is “In Figure 2B”? it should be Figure 3B. Please check these mistakes throughout the manuscript and accurately address the figures. There are some other cases in the manuscript.
18- From Figure 5C there is a clear aggregation of SrO nanoparticles in the film even at low concentration which is 10 wt.%. This means that the nanoparticles are not uniformly distributed in the films, and this effect on the overall properties of the films. The homogeneity of the films should be further discussed and argued with providing logical reasoning?
19- In page 6 line 220 what is the minus sign in Debye-Scherrer formula?
20- Lastly why authors ended up using those concentrations (5, 10, 15, 20 wt.%) of SrO? What will be the outcomes if slightly lower concentration was used? This is a random selection or based on a scientific reason?
Author Response
Responses to the reviewer comments
Authors would like to thank the reviewer for his valuable comments.
Please see the following table of our responses, corrections and explanation based on the reviewer comments.
|
Reviewer 2 Comments |
Our Reply |
1 |
All abbreviations should be first defined then used throughout the manuscript. For example, SrO in “Abstract” section, or CNS depressing (page 2, line 61) and some other cases in the manuscript. |
Thank you for the reviewer comment, the abbreviations are revised throughout the manuscript. |
2 |
In the introduction section the authors need to put more effort to clearly address the gap in the current researches in this area and why carrying out this research is important. In other words, what is the research questions the authors have? I feel the authors should clearly state what are the reasons for performing the study. They can add this to the last paragraph of the introduction. Furthermore, the third paragraph (page 2, line 59) is not well-structured, and there is no topic sentence that should give the reader a clue about the entire paragraph. Thus, it should be modified and expanded with more recent works from the literature. |
Additional transition sentences are added in the introduction part. Also, the aim of the study is clarified. In addition, more citations are added to the introduction part. |
3 |
Figure 1 is placed in the introduction section without been addressed or explained in the text. This should be fixed. |
Thank you very much for your notice, the figure is addressed in the text. |
4 |
In Materials and Methods section, the molecular weight of used raw material like (CS) should be added, as it significantly impacts on the overall behavior of used host medium. |
Because it is polymer with an average molecular weight, Sigma-Aldrich provided us that it is medium molecular weight with the density value (C3646, density = 0.15–0.3 g/cm3), anyhow, the medium molecular weight is mentioned in materials section. Thanks
|
5 |
In Materials and Methods section the range of FTIR is specified to be (400 – 4,000 cm−1) while in the section (3.2.1. Characterization via FTIR) (page 5) the range is different and it should be corrected. Similar mistake should be corrected for the XRD were the range is mentioned to be 2θ of 10° and 80°; while in Figure 5 the range is from 0° and 60° and this should be corrected. |
Thank you very much, the ranges are corrected as measured in figures. |
6 |
In page 5 line 88, what is the meaning of 4nm Pt? This should be clarified. |
It means that the samples were coated with 4 nm thickness of platinum layer. Thank you, the sentence is clarified. |
7 |
All used equations should be numbered in sequence and cited (such equation in page 6 line 220). |
The equation is numbered and cited in the text. |
8 |
In page 3 line 96 authors stated that “the viscous solution was filtered” This statement needs to be more clarified by addressing the filtering process and what has been used as a filter. |
A description for the filtering process is added. |
9 |
There are some grammatical, spelling, format and punctuation mistakes, please revise the manuscript accurately (for instant page 3 line 100 the word “diffeerent”, and some other cases in the manuscript. |
The manuscript is revised and English editing by Journal editorial board will be done. |
10 |
What is the purpose of soaking the films back in distill water after film formation, as stated in page 3 line 103? How author confirm that there is no dissolution of the film during this process? This should be clarified. |
Soaking the film in distilled water to wash it form any salts that are formed by neutralization of acetic acid with the base. The film, after neutralization, is not soluble in water so can be used as heterogeneous catalyst in the investigated reactions. |
11 |
Some citation colors are different and not accurately include the information mentioned in the sentences; for instant, page 4 line (163, 173), page 5 line (188), page 6 line (219). |
Thank you for your comment, - Page 4, line 163; actually, it is used previously and modified in this paper using microwave irradiation. The sentence is revised. - Page 5, line 188; we only refer to the characteristic peaks of chitosan as we usually obtained in our previous work. - Page 6, line 219; the reference is related to the first report about Debye-Scherrer equation and its use. |
12 |
Section (3.1. Preparation of chitosan-strontium oxide nanocomposite film) should be moved to the Material and Method section, then all the section should be numbered in sequence. |
The preparation of chitosan is moved to experimental part and all section are re-numbered as the reviewer suggestion. Thanks for the valuable notice. |
13 |
In page 4 line 168 “Figure 1 shows” should be changed to Figure 2 as the FTIR is presented in Figure 2. |
Figure 1 is corrected in the text to be Figure 2. |
14 |
The citation style should be unified as the one written in page 4 line 171 [(Cardenas and Miranda, 2004)]. |
Thank you very much, the sentence is revised and correct. |
15 |
In page 4 line 172 it is written u = 1658 cm-1? What is u? It should be defined. |
Sorry, the style is changed it is corrected in the text. It is the wave number u = 1658 and 1609 cm-1 |
16 |
In Figure 2 the FTIR spectra for all other concentration of SrO should be added in order to clarify the impact of nanoparticle concentration on the structural behavior of the prepared thin films. Similarly, the SEM and XRD for all other concentration are necessary to be added. |
Thank you for your comment. In FTIR and SEM, there is no clear changes with changing the SrO concentration. and after examining the catalytic activity of the different concentrations, we found that the best concentration was 10% wt. that is why we considered this concentration in all illustrations as usual in all of our previous publications. |
17 |
In page 5 line 188, what is “In Figure 2B”? it should be Figure 3B. Please check these mistakes throughout the manuscript and accurately address the figures. There are some other cases in the manuscript. |
Figure 2B corrected to be Figure 3B Thanks, all the numbers of the figures are checked and revised.
|
18 |
From Figure 5C there is a clear aggregation of SrO nanoparticles in the film even at low concentration which is 10 wt.%. This means that the nanoparticles are not uniformly distributed in the films, and this effect on the overall properties of the films. The homogeneity of the films should be further discussed and argued with providing logical reasoning? |
You mean Figure 3C, you are right, the figure is not selected correctly so, another image is provided instead of this view. |
19 |
In page 6 line 220 what is the minus sign in Debye-Scherrer formula? |
The equation is corrected. thanks |
20 |
Lastly why authors ended up using those concentrations (5, 10, 15, 20 wt.%) of SrO? What will be the outcomes if slightly lower concentration was used? This is a random selection or based on a scientific reason? |
Based on our preliminary tests, the investigated catalytic reactions could be conducted by 10-20% wt. of the unrecovered SrO (relative to the substrate), thus in this study we tried to use a similar wt% but within the chitosan matrix (as it used as recoverable heterogeneous catalyst). Thanks for valuable comments |

Round 2
Reviewer 1 Report
The authors have addressed most of my concerns. I still have a few comments:
1. Please add the NMR spectra of 5a to the manuscript or to the supporting information.
2. I suggested the authors to add higher magnification SEM images, not higher resolution image. Figure 1 shows that in principle there should still be spherical nanoparticles embedded in the chitosan matrix. I would think the size distribution and morphology of nanoparticles could affect the catalytic properties of the resulting nanocomposites, and Scherrer equation itself may not be enough to give us such information. Figure 3c only shows large aggregates, can the authors comment on the size distribution or morphological features of CS-SrO nanocomposites they synthesized?
Author Response
|
Reviewer 1 Comments |
Our Reply |
1 |
Please add the NMR spectra of 5a to the manuscript or to the supporting information. |
Analytical data for compound 5a is added in supporting information. |
2 |
I suggested the authors to add higher magnification SEM images, not higher resolution image. Figure 1 shows that in principle there should still be spherical nanoparticles embedded in the chitosan matrix. I would think the size distribution and morphology of nanoparticles could affect the catalytic properties of the resulting nanocomposites, and Scherrer equation itself may not be enough to give us such information. Figure 3c only shows large aggregates, can the authors comment on the size distribution or morphological features of CS-SrO nanocomposites they synthesized? |
SEM image of higher magnification and other analytical data for the CS-SrO nanocomposite are added in the supporting information. |

Reviewer 2 Report
Dear Authors,
Thanks for considering the proposed suggestions and comments. The manuscript is improved however these two points should be corrected
· The FTIR range is still not corrected, in the figure the range is from 500-4000 cm-1, while in experimental section authors stated that it is from 0-4000 cm-1. Also the XRD in Figure 5 is from 5-60o not 10-60o as stated in the experimental section. This is second time I comment on this I do not understand why authors do not pay attention!
* Newly added Figure 3C should be labeled.
Author Response
|
Reviewer 2 Comments |
Our Reply |
1 |
The FTIR range is still not corrected, in the figure the range is from 500-4000 cm-1, while in experimental section authors stated that it is from 0-4000 cm-1. Also the XRD in Figure 5 is from 5-60o not 10-60o as stated in the experimental section. This is second time I comment on this I do not understand why authors do not pay attention!
|
Thank you for your comment, the scales are corrected. |
2 |
Newly added Figure 3C should be labeled. |
Thank you for your comment, the Fig. 3C is labelled. |